# Immunostaining protocol for infiltrating brain cancer spheroids for light-sheet imaging

**Benedicte Bjørknes[1], Oliver Emil Neye[1], Petra Hamerlik[2¤], Liselotte Jauffred[1]***

**1** The Niels Bohr Institute, University of Copenhagen, Copenhagen, Denmark, **2** Danish Cancer Society, Copenhagen, Denmark

¤ Current address: Division of Cancer Sciences, University of Manchester, Manchester, United Kingdom
* jauffred@nbi.dk

## Abstract

Glioblastoma tumors form in brains' white matter and are fast-growing and aggressive. Poor prognosis is the result of therapeutic resistance and infiltrating growth into the surrounding brain. Here we present a protocol for the detection of the cytoskeleton intermediate filament, vimentin, in cells at the proliferating spheroid surface. By combining a classical invasion assay with immunofluorescence and light-sheet imaging, we find that it is exactly these cytoskeleton-reinforcing cells on the spheroid's surface that will start the infiltration. We anticipate our results to be the starting point of more sophisticated investigation of anti-cancer drug effects on cytoskeleton reorganisation.

## Introduction

Tumor spheroids of primary or xenograft cell lines are well-characterized 3D models of cancer; mimicking essential features of tumors and their micro-environment, such as gradients of oxygen, metabolites, and pH. So when cancer cells grow as spheroids—instead of monolayers—we find an outer proliferating zone, an intermediate quiescent region with limited oxygen and metabolites [1, 2], and occasionally even a necrotic core [3]. This strong cellular and molecular heterogeneity in spheroids makes them adequate models of cancer.

Under normal physiological conditions—when cells are proliferating in tumors—cell-cell adhesion and tissue formation is favored. In contrast, a hallmark of glioblastoma tumors is the infiltrating growth [4], resulting from mesenchymal tumor cells acquiring migrating phenotypes [5, 6].

This phenotypic switch is associated with major rearrangement of the cytoskeleton. For instance in invasive glioblastoma cells, by overexpression of the class III intermediate filament protein vimentin (vim). While vim might not be directly involved in motility, the filaments add mechanical strength through enforcement of the cytoskeleton [7] in invasive phenotypes of both mesenchymal and epithelial cancers [8]. Interestingly, vim expression enforces cell motility even without down-regulating, e.g., cadherins. Thus, indicating that vim expression is sufficient for epithelial cell migration (see Ref. [9] for a review).

Although we understand the gain of invasive traits in individual cells, we can often only guess what the effects will be on larger tumor settings. Therefore, with this protocol we

**Data Availability Statement:** All relevant data are within the paper and its Supporting information files.

**Funding:** LJ:The Danish National Research Foundation (0165-00032B, 0165-00103B) Danish

National Research Councils (DNRF116) Novo Nordisk Foundation (NNF14OC0011361) The funders had no role in study design, data collection and analysis, decision to publish, or preparation of the manuscript.

integrate recent advances within 3D primary spheroid culture with the progress of Airy beam light-sheet fluorescence imaging (LSM). We use a combination of nuclear stain and vim antibody marker to show how differently surface-attached human glioblastoma cells (2D) grow, compared to cells in spheroids (3D). Specifically, we find that vim expression is uniform in 2D and limited to the outermost cell-layer of the spheroid in 3D. This is the offset for presenting a protocol for antibody-staining in an invasion assays, which requires staining through the extra-cellular matrix. The expected result, is precise mapping of cytoskeleton markers in infiltrating finger-like protrusions originating from the spheroid surface. We anticipate that the synthesis of these methods can provide us a conceptual model of spatial regulation of invasion to point out new directions to encompass the infiltrating growth of glioblastoma cancers.

## Materials and methods

The protocol described in this peer-reviewed article is published on protocols.io, https://dx.doi.org/10.17504/protocols.io.eq2ly77krlx9/v2 and is included for printing as S1 File with this article.

### Ethics statement

Glioblastoma (GBM) cell cultures were established from freshly resected tumor tissue including informed consent from each patient, as outlined by the Regional Danish Ethical Committee/Danish Data Protection Agency (H-3–2009-136-63114). Written consent has been acquired from participants whose tissue was used to generate our models. GBM lines were passaged as xenografts in the subcutaneous flank of NOG mice (Taconic, TAC:nog) according to Danish Welfare Law on Animal Experiments Act no 1306, protocol: 2012–15-2934-00636.

### Image processing

FIJI [10] was used for initial processing, pseudo-coloring etc. To examine radial distributions of vim and DNA in 3D light-sheet images, the following image was constructed: The spheroid closest surface was identified and a 4 $\mu$m thick slice (10 images) in 50 $\mu$m into the spheroid was collected. From this slice an average projection was constructed. Then with a custom-build Matlab procedure, the intensity is averaged over all pixels in a ring in a specific radial distance, as detailed in [11] and references therein. This corresponds to averaging over a rotating radial vector, as sketched in the inset of Fig 3. This way, the inside of the spheroid can be studied, as well as the spheroid's surface. Even though the spheroids are seeded with the same number of cells the size varied, so to compare the expression in spheroids with different sizes, the intensity were plotted from the edge of each spheroid. The resulting arrays of intensities versus spheroid radii was translated into arrays of intensities versus distance from surface, $d$, to make results independent of size variations. Thus, we define $d = 0$ to be the point where the nuclear stain (DNA) intensities are falling off most rapidly, i.e., where the magnitude of the time-derivative is the largest. Afterwards, all data is translated accordingly.

## Expected results

Our model system is the human glioma xenograft T115 complemented with the primary cell line U87. Through mapping of protein expression, i.e., immunofluorescence, we investigated how glioblastoma cells' fate depend on spatial organisation. Specifically, we searched for differences in expression levels, when grown in either 2D or 3D configurations.

## Surface-attached cells show homogeneous marker expression

After verifying the specificity of our immunological setup (see the Materials and methods for details), we seeded T115 cells in glass-bottom culture dishes and after 3 days of incubation, the cells were fixed and stained with the set of vim antibodies and nuclear stain.

In surface-attached T115, we detected vim (yellow) and cell nuclei (blue), see the upper panel of Fig 1. As the xenograft cell line naturally grow like spheroids when seeded in medium, we anticipate that they attach to the glass surface of the culture dish more loosely. In accordance with our finding that only few cells started to form clusters attached to the surface. So, whilst the staining procedure includes multiple washing, all loosely attached cells has been washed away. Therefore, we suspect that this result favors a sub-population of the highly heterogeneous cell population with specific adherent properties.

The lower panel of Fig 1 shows similar detection in surface-attached U87 cells, in accordance with earlier findings [12] and as expected for this mesenchymal cell type.

## Mesenchymal marker expression in 2D

In aim of investigating how geometry affects the expression of mesenchymal markers, we sat up a wound healing (or scratch) assay. In brief, we prepared two patches of surface-attached U87 cell monolayers (as in Fig 1) but separated by a 500 $\mu$m gap. Then, after 24 hours of incubation, cells were fixed and stained. We found that U87 cells migrated to close the gap, i.e., healing the wound, as shown in Fig 2. We found that the vim expression level was homogeneous and independent of the distance to the gap in this 2D geometry.

## Strong gradient of mesenchymal marker expression in 3D

It is well-established that the organization of surface-attached cancer cells is very different from the organization in a dense tumor-like structure, i.e., a spheroid. Therefore, we grew

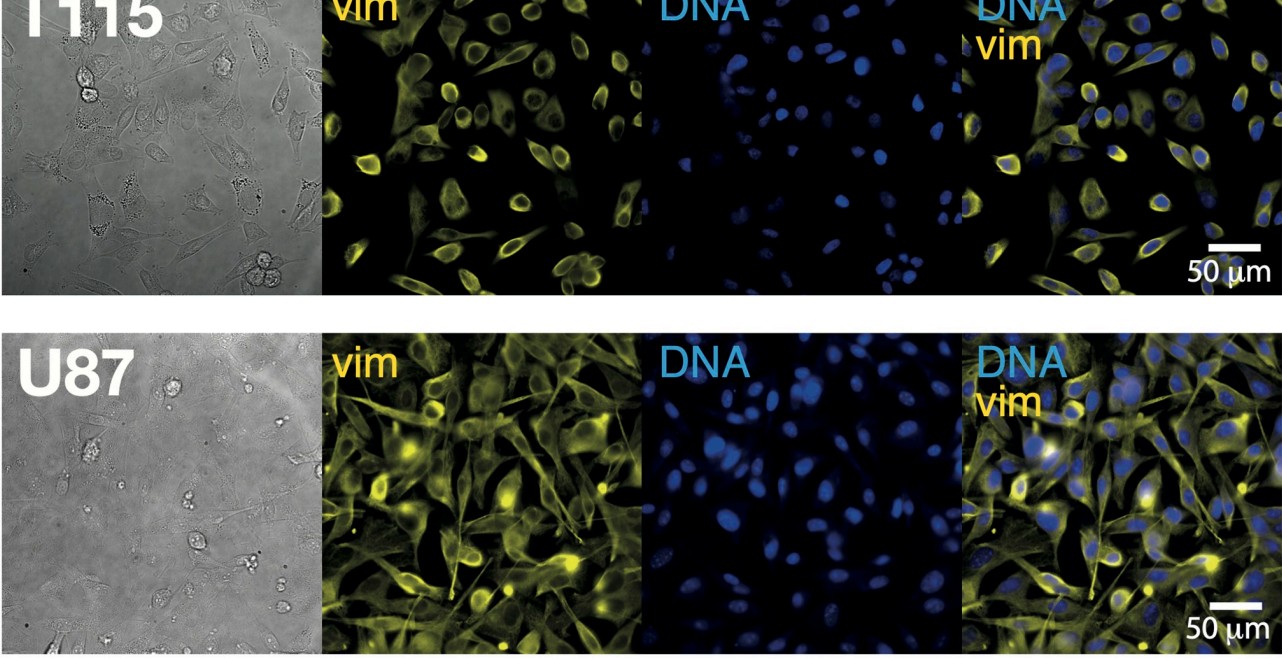

**Fig 1. Vim marker expression in brain cancer cells.** Immunofluorescence in parallel assays of vim (yellow) and cell nuclei identified by a nuclear stain (blue) in both GBM xenograft cell line T115 and cell line U87.

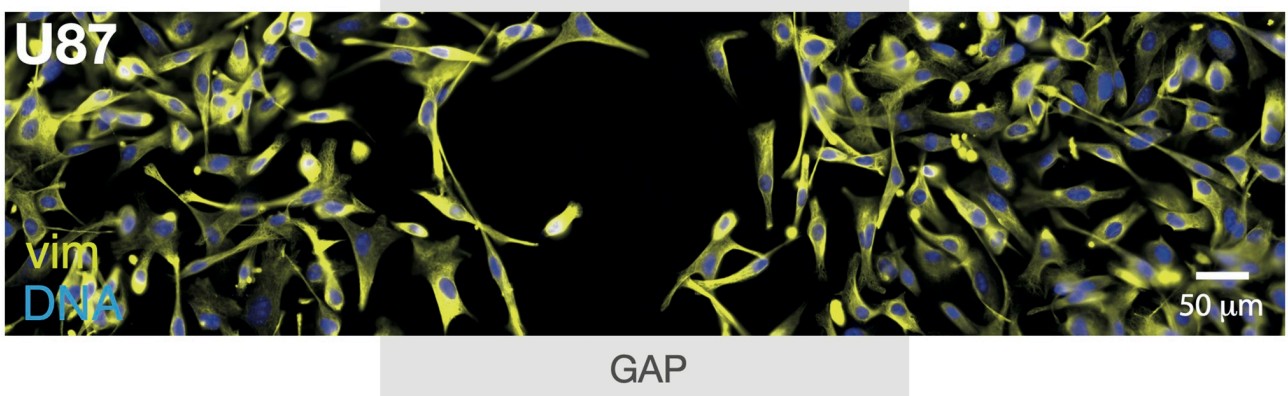

**Fig 2. Spatial patterns of vim marker in 2D.** Wound healing experiment in surface-attached U87 brain cancer cells, where the original wound is indicated by the light grey area (GAP). Immunofluorescence parallel assays after 24 hours growth of vim (yellow) and cell nuclei identified by a nuclear stain (blue).

glioma spheroids by seeding 500 cells in a U-formed well coated with a covalently bound hydrogel layer to inhibit cell attachments and incubated them for 3 days before fixing and staining. For imaging we used a two-laser LSM, where vim were co-stained with nuclear stain (DNA) to compare penetration depth of the stains and better separate the cells in the spheroids. As seen from Fig 3A and the merged channels in Fig 3B (region of interest from Fig 3A), we found high, uniform vim expression on the spheroid surface. One advantage of LSM is that despite the dense spheroid environment, we can still image cells placed several cell layers below the surface without using clearing agents. So even though the maximum intensity projection (Fig 3B) show uniform vim expression on the spheroid surface, the expression along the radial vector is strongly non-uniform. This is seen from the expression levels of DNA and vim versus distance from surface, $d$, in Fig 3C. Strikingly, vim expression is highly localized to

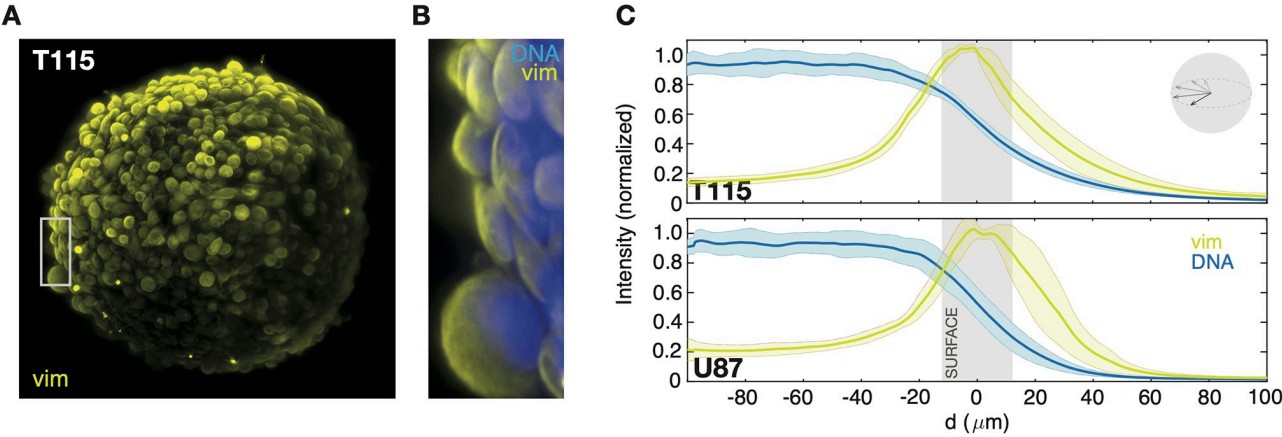

**Fig 3. Spatial patterns of markers in 3D. A:** Natural spheroid formation primary brain cancer (T115) in liquid cell culture. Maximum intensity z-projection of immunofluorescent vim (yellow). **B:** Merged image of the region of interest in (B) with immunofluorescent vim (yellow) and nuclei stain (blue). **C:** Radial distribution of vim (yellow) and nuclei stain (DNA, blue) versus the distance from the spheroid's surface, $d$, for both cell lines. The full line is the mean immunostain intensities averaged over different spheroids (N = 4), the shaded areas designate one standard deviation, and the grey vertical bar (SURFACE) indicates the roughness of the spheroid's surface; corresponding to 1–2 cell widths. Inset: Sketch of how the intensity distribution over the radial vector is averaged. Intensities are measured for a rotating radial vector in a 4 $\mu m$ slice (10 images) of the spheroid (50 $\mu m$ depth) as detailed in Materials and Methods.

the surface, whilst DNA basal levels are uniform within the spheroid. Furthermore, vim is mainly found close to the nuclear envelope and extend toward the cell periphery, as earlier reported for intermediate filaments in other non-motile tumor cell types [13]. To check that this result is not biased by penetration issues of the antibodies, we tried different reaction times (from 1 hour to 3 hours). We found no obvious differences, so we anticipate that the marker gradients are independent of incubation times. Furthermore, as DNA was stained at low homogeneous levels penetration does not seem to limit at shallow spheroid depths ($< 100\ \mu$m).

This strong vim gradient fits in a view, where surface cells are most prone to start invasion of the surrounding tissue. One possible mechanism is that it is the interaction with the local environment that induces the phenotypic response.

### Mesenchymal marker expression in invading spheroids

To verify the assumption that invasion will initiate from the vim-expressing surface cells, we prepared an invasion assay: 3 days old spheroids were embedded in the mammalian cellular basement matrix Matrigel™ mixed 1:1 with medium. Hereafter, the spheroids where incubated for 24 hours, which allowed them to invade the matrix before fixation and staining.

As the extracellular matrix is a source of high mechanical force, spheroids respond to the mechanotransduction signals by starting invasion of the surrounding matrix, as seen from Fig 4A. When zooming in as done in Fig 4B (region of interest in Fig 4A), we find long finger-like protrusions—extending from the outermost layer of cells—that are highly vim-enforced. In accordance with earlier findings, where vim was found to elongate in parallel with the

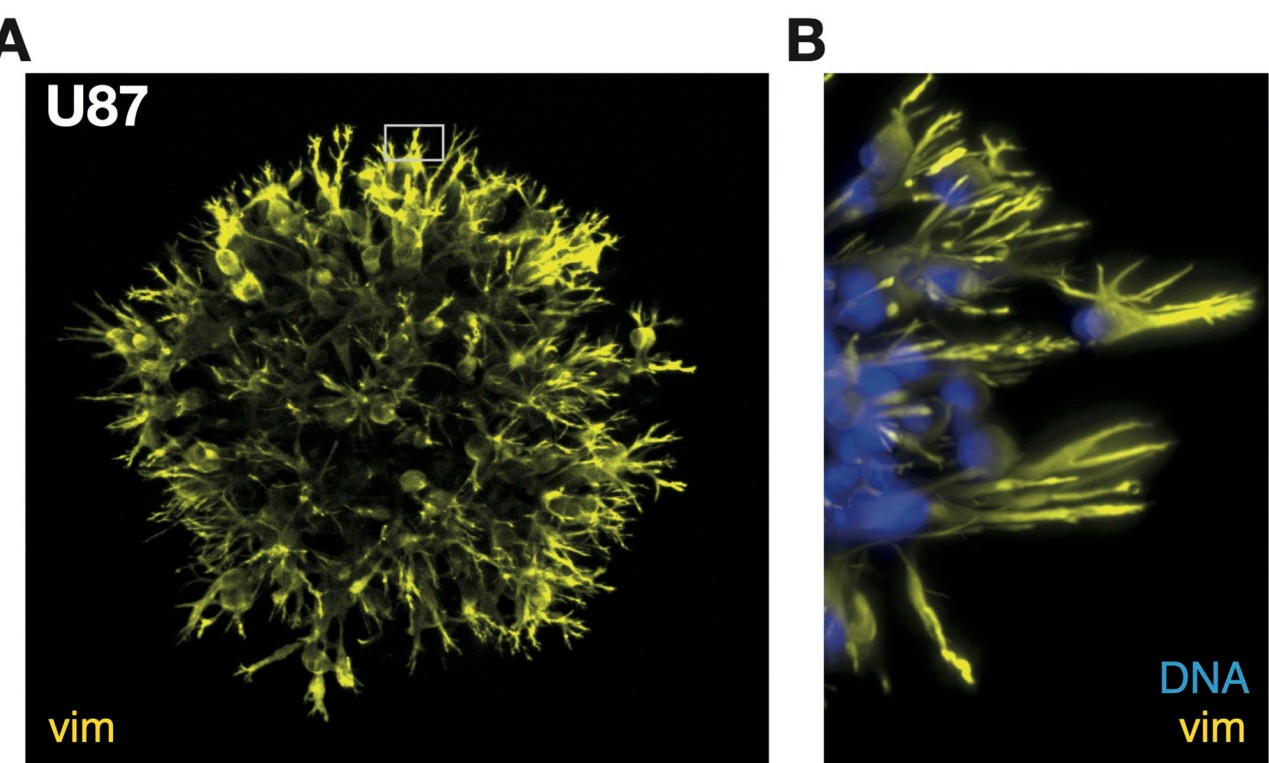

**Fig 4. Invasion assay. A:** Initiation of invasion for glioblastoma U87in a brain-mimicking matrix. Maximum intensity z-projection of immunofluorescent vim (yellow).**B:** Merged image of the region of interest in (B) with immunofluorescent vim (yellow) and nuclei stain (blue).

microtubules network towards focal adhesions in the filopodia [14]. This result fits with the proposed view that the infiltrations arising from brain cancer are formed on the spheroid surface.

## Discussion

As glioblastomas are of mesenchymal origin, their dominant intermediate filament is the cytoskeleton structural protein (vim) [15–17], which is associated with increased proliferation and invasion [8]. Even though xenograft glioblastoma cells (T115) is naturally differentiated and with subpopulations of different characteristics, we found vim to be homogeneously over-expressed on the spheroid surface. In contrast, previous studies have shown that ephitelial cadherin (E-cad) is rarely expressed in glioma cells [6, 15, 18]. Very interestingly, there are also other experimental evidence that the expression levels of the mesenchymal markers are higher at the rim of a spheroid compared to the shielded interior and vice-versa for epithelial markers [19, 20].

When embedded in an extracellular matrix, the cells on the proliferating surface begun invasion by long finger-like protrusions, which is a sign of collective migratory behavior through gained motility and retained cell-cell adhesion [4]. This is in accordance with earlier findings for similar spheroids (U87) [3, 21–23] as well as for other glioblastoma spheroids [3], glioma spheroids [24] etc. Taking these results together, invasion of the micro-environment is initiated from the tumors' surface cells. We speculate that when the first cells leave the spheroid surface, they expose the cells underneath, and the transition will be triggered to reinforce the cytoskeleton and infiltrate the surroundings.

This infiltrating nature makes it fairly difficult to remove all cancerous tissue by surgery, thus leading to tumor recurrence from margins of resection cavities. Furthermore, various therapies promote infiltration through the expression of growth factors and enrichment of neural and glioma stem cells [25, 26]. It is therefore particularly important for our understanding of brain cancer therapy resistance to unravel the mechanism of infiltrating growth.

## Supporting information

**S1 File. Step-by-step protocol, also available on protocols.io.**
(PDF)

## Author Contributions

**Conceptualization:** Petra Hamerlik.

**Data curation:** Liselotte Jauffred.

**Formal analysis:** Benedicte Bjørknes.

**Funding acquisition:** Liselotte Jauffred.

**Investigation:** Benedicte Bjørknes, Oliver Emil Neye.

**Methodology:** Benedicte Bjørknes, Oliver Emil Neye, Petra Hamerlik.

**Project administration:** Petra Hamerlik, Liselotte Jauffred.

**Supervision:** Petra Hamerlik, Liselotte Jauffred.

**Visualization:** Benedicte Bjørknes, Oliver Emil Neye.

**Writing – original draft:** Benedicte Bjørknes, Oliver Emil Neye, Petra Hamerlik, Liselotte Jauffred.

**Writing – review & editing:** Benedicte Bjørknes, Oliver Emil Neye, Petra Hamerlik, Liselotte Jauffred.

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
