## [Decision Letter · Decision Letter 0]

20 Dec 2022

PONE-D-22-31029Immunostaining protocol for infiltrating brain cancer spheroids for light-sheet imagingPLOS ONE

Dear Dr. Jauffred,

Thank you for submitting your manuscript to PLOS ONE. After careful consideration, we feel that it has merit but does not fully meet PLOS ONE’s publication criteria as it currently stands. Therefore, we invite you to submit a revised version of the manuscript that addresses the points raised during the review process.

We look forward to receiving your revised manuscript.

Kind regards,

Michelle R. Dawson

Section Editor

PLOS ONE

Journal Requirements:

3. Thank you for stating the following financial disclosure: "NO"

6. We note you have not yet provided a protocols.io PDF version of your protocol and/or a protocols.io DOI. When you submit your revision, please provide a PDF version of your protocol as generated by protocols.io (the file will have the protocols.io logo in the upper right corner of the first page) as a Supporting Information file. The filename should be S1_file.pdf, and you should enter “S1 File” into the Description field. Any additional protocols should be numbered S2, S3, and so on. 

Please also follow the instructions for Supporting Information captions [https://journals.plos.org/plosone/s/supporting-information#loc-captions]. The title in the caption should read: “Step-by-step protocol, also available on protocols.io.”

Please assign your protocol a protocols.io DOI, if you have not already done so, and include the following line in the Materials and Methods section of your manuscript: “The protocol described in this peer-reviewed article is published on protocols.io (https://dx.doi.org/10.17504/protocols.io.[...]) and is included for printing purposes as S1 File.” You should also supply the DOI in the Protocols.io DOI field of the submission form when you submit your revision.

If you have not yet uploaded your protocol to protocols.io, you are invited to use the platform’s protocol entry service [https://www.protocols.io/we-enter-protocols] for doing so, at no charge. Through this service, the team at protocols.io will enter your protocol for you and format it in a way that takes advantage of the platform’s features. 

When submitting your protocol to the protocol entry service please include the customer code PLOS2022 in the Note field and indicate that your protocol is associated with a PLOS ONE Lab Protocol Submission. You should also include the title and manuscript number of your PLOS ONE submission.

**Additional Editor Comments:**

Please revise your manuscript to address the reviewers comments. Also, please review your submission carefully to eliminate any typos or grammatical errors.

I look forward to receiving your revised manuscript.

Reviewers' comments:

Reviewer's Responses to Questions

**Comments to the Author**

1. Does the manuscript report a protocol which is of utility to the research community and adds value to the published literature?

Reviewer #1: Yes

Reviewer #2: Yes

2. Has the protocol been described in sufficient detail?

To answer this question, please click the link to protocols.io in the Materials and Methods section of the manuscript (if a link has been provided) or consult the step-by-step protocol in the Supporting Information files.

The step-by-step protocol should contain sufficient detail for another researcher to be able to reproduce all experiments and analyses.

Reviewer #1: Yes

Reviewer #2: Partly

3. Does the protocol describe a validated method?

Reviewer #1: Yes

Reviewer #2: Yes

4. If the manuscript contains new data, have the authors made this data fully available?

Reviewer #1: Yes

Reviewer #2: Yes

**5. Is the article presented in an intelligible fashion and written in standard English?**

Reviewer #1: Yes

Reviewer #2: Yes

6. Review Comments to the Author

Reviewer #1: The submitted manuscript describes a very useful technique for staining and imaging infiltrating brain cancer cells using a spheroid model. It is easy to imagine that this technique could be utilized for models of various types of infiltrating tumors. It would be useful, but may not be strictly necessary to have some quantifiable benchmarks associated with the protocol - what percentage of surface cells were stained using this method, and what percentage of those cells are expected to infiltrate the Matrigel?

The speculation that as the cells migrate from the tumor surface, the cells underneath will then repeat the process of vim expression and infiltration is an interesting one, but is this something that the authors have been able to validate in any way? Would this require another round of staining or is this something that could be easily observed in the same assay based on the penetration of the vim staining within the spheroid? An answer to this question could be rather impactful in terms of the application of this technique.

The manuscript could use some slight editing for grammar and the protocol has some additional characters (????) that don't influence the method but add some confusion.

Reviewer #2: This manuscript described a protocol for immunostaining of glioblastoma spheroid for two laser light sheet imaging and proved the utility of this protocol by showing a distinct surface staining of vimentin on spheroid and the protrusion of vimentin in migration assay. However, as a protocol article, the current version didn't provide the readers enough detail of the experiment, and there are some confusions in the protocol. I highly recommend the authors carefully organize the protocol part of this article make it really useful to the community. And my suggestions are as follows.

1. what are those question markers in section "cell culture of primary specimen"

2. Replace all the "agent" to " reagent"

3. in the fixation reagent section there is description about tips of how to wash cells. Where in the Procedure section these tips should be applied to? Is in the "Immunostaining spheroids through hydrogel"? However, there are different description in the "Immunostaining spheroids through hydrogel" fixation section. Which one the readers should follow? It is really confusing!

4. in the Cell Culture section, please specify the number of cells and size of culture dishes. Not just use the description"small culture plate"

5. In Gravitation-assisted spheroid formation what is the sentence "The spheroids can be immunofluorescence stained directly using the procedure in section Immunostaining surface-attached

cells" meaning?

6. In "Imaging spheroids in hydrogel", why the authors put plate on ice to liquefy the hydrogel matrix, isn't matrigel already dissolved in PFA? And can the authors please add a schematic cartoon to show how to set up the imaging tube?

7. In "Immunostaining surface-attached cells", why wash the cells with cold PBS, doesn't that detach the cells? And please specify the time of washes.

8. For primary antibody staining, what time and temperature of incubation the author was using for Vim1? Because in the data part, authors mentioned they have tried 1 hour and 3 hour and no difference was observed. Please make it clear in the protocol and make sure consistent with the data description.

9. For the Data part, in figure 3C, it is hard to understand the way of quantification of the intensity based on the schematic the authors provided.

7. PLOS authors have the option to publish the peer review history of their article (what does this mean?). If published, this will include your full peer review and any attached files.

Reviewer #1: No

Reviewer #2: No

---

## [Editor Report · Decision Letter 1]

11 Jan 2023

PONE-D-22-31029R1Immunostaining protocol for infiltrating brain cancer spheroids for light-sheet imagingPLOS ONE

Dear Dr. Jauffred,

Thank you for submitting your manuscript to PLOS ONE. After careful consideration, we feel that it has merit but does not fully meet PLOS ONE’s publication criteria as it currently stands. Therefore, we invite you to submit a revised version of the manuscript that addresses the points raised during the review process.

We look forward to receiving your revised manuscript.

Kind regards,

Michelle R. Dawson

Section Editor

PLOS ONE

Journal Requirements:

Additional Editor Comments (if provided):

Thank you for addressing the main concerns in the reviews. A few minor concerns remain. First, in the description of isolated cancer cells it is unclear whether this is human subjects research or not, if it is human subjects research an IRB approval would be required and that should be stated. Otherwise, the author should use the appropriate language to show that is not human subjects research. In addition, the reviewers had some concerns with the grammar and writing. I noticed some portions of this protocol are still very conversational. Please read it once more and correct any colloquial language since this is a scientific paper. After these small issues are addressed, I would be happy to accept your protocol.
---

## [Editor Report · Decision Letter 2]

17 Jan 2023

Immunostaining protocol for infiltrating brain cancer spheroids for light-sheet imaging

PONE-D-22-31029R2

Dear Dr. Jauffred,

We’re pleased to inform you that your manuscript has been judged scientifically suitable for publication and will be formally accepted for publication once it meets all outstanding technical requirements.

Kind regards,

Michelle R. Dawson

Section Editor

PLOS ONE

---

## [Editor Report · Acceptance letter]

31 Jan 2023

PONE-D-22-31029R2 

Immunostaining protocol for infiltrating brain cancer spheroids for light-sheet imaging 

Dear Dr. Jauffred:

I'm pleased to inform you that your manuscript has been deemed suitable for publication in PLOS ONE. Congratulations! Your manuscript is now with our production department. 

Kind regards, 

on behalf of

Dr. Michelle R. Dawson 

Section Editor

PLOS ONE